# INFUSING LATTICE SYMMETRY PRIORS IN NEURAL NETWORKS USING SOFT ATTENTION MASKS

## ABSTRACT

Infusing inductive biases and knowledge priors in artificial neural networks is a promising approach for achieving sample efficiency in current deep learning models. Core knowledge priors of human intelligence have been studied extensively in developmental science and recent work has postulated the idea that research on artificial intelligence should revolve around the same basic priors. As a step towards this direction, in this paper, we introduce LATFORMER, a model that incorporates lattice geometry and topology priors in attention masks. Our study of the properties of these masks motivates a modification to the standard attention mechanism, where attention weights are scaled using soft attention masks generated by a convolutional neural network. Our experiments on ARC and on synthetic visual reasoning tasks show that LATFORMER requires 2-orders of magnitude fewer data than standard attention and transformers in these tasks. Moreover, our results on ARC tasks that incorporate geometric priors provide preliminary evidence that deep learning can tackle this complex dataset, which is widely viewed as an important open challenge for AI research.

## 1 INTRODUCTION

Infusing inductive biases and knowledge priors in neural networks is regarded as a critical step to improve their sample efficiency (Battaglia et al., 2018; Bengio, 2017; Lake et al., 2017; Lake & Baroni, 2018; Bahdanau et al., 2019). The *Core Knowledge* priors for human intelligence have been studied extensively in developmental science (Spelke & Kinzler, 2007), following the theory that humans are endowed with a small number of separable systems of core knowledge, so that new flexible skills and belief systems can build on these core foundations. Recent research in artificial intelligence (AI) has postulated the idea that the same priors should be incorporated in AI systems (Chollet, 2019), but it is an open question how to incorporate these priors in neural networks.

Following this chain of thought, the Abstraction and Reasoning Corpus (ARC) (Chollet, 2019) was proposed as an AI benchmark built on top of the Core Knowledge priors from developmental science. Chollet (2019) posits that ARC "*cannot be meaningfully approached by current machine learning techniques, including Deep Learning*". Further, he argues that developing a domain-specific approach based on the Core Knowledge priors is a challenging first step and that "*solving this specific subproblem is critical to general AI progress*".

An important category of Core Knowledge priors includes *geometry and topology priors*. Indeed, significant attention has been devoted to incorporating such priors in deep learning architectures by rendering neural networks invariant (or equivariant) to transformations represented through group actions (Bronstein et al., 2021). Group invariant learning helps to build models that systematically *ignore* specific transformations applied to the input (such as translations or rotations).

We take a complementary perspective and aim to help neural networks to learn functions that incorporate geometric transformations of their input (rather than to be invariant to such transformations). In particular, we focus on group actions that belong to the symmetry group of a lattice. These transformations are pervasive in machine learning applications, as basic transformations of sequences, images, and other higher-dimensional regular grids fall in this category. While attention and transformers can in principle learn these kind of group actions, we show that they require a significant amount of training data to do so.

Figure 1: We consider problems that involve learning a geometric transformation on the input data as a sub-problem. The displayed task (taken from ARC) entails learning to map, for each pair, the left to the right image. We investigate how to solve such tasks more sample-efficiently by imbuing self-attention with the ability to exploit lattice symmetry priors.

To address this sample complexity issue, we introduce LATFORMER, a model that relies on attention masks in order to learn actions belonging to the symmetry group of a lattice, such as translation, rotation, reflection, and scaling, in a *differentiable* manner. We show that, for any such action, there exists an attention mask such that an untrained self-attention mechanism initialized to the identity function performs that action. We further prove that these attention masks can be expressed as convolutions of the identity, which motivates a modification to the standard attention module where the attention weights are modulated by a mask generated by a convolutional neural network (CNN).

Our paper focuses on ARC and its variants. We see the extension of LATFORMER to other tasks as a promising avenue for future research. Therefore, we conducted an evaluation of our approach based on synthetic tasks, ARC and the recently proposed *LARC* (Acquaviva et al., 2021). First, to probe the sample efficiency of our method, we compared its ability to learn synthetic geometric transformations against Transformers and attention modules. Then, we annotated ARC tasks based on the knowledge priors they require, and we evaluated LATFORMER on the ARC and LARC tasks requiring geometric knowledge priors. Our results provide evidence that LATFORMER can learn geometric transformations with 2 orders of magnitude fewer training data than transformers and attention. We also provide the first neural network reaching good performance on a subset of ARC, suggesting that this kind of problem does not lie out of the reach of deep learning models.

## 2 FORMALIZING THE GROUP-ACTION LEARNING PROBLEM

We are interested in helping neural networks to learn lattice transformations sample efficiently by infusing knowledge priors in the model. Motivated by ARC, we focus on learning geometric transformations that belong to the symmetry group of a lattice. This pertains to the more general problem of learning group actions given the input and the output of the transformation.

Concretely, we consider input-output transformations involving a group element g taken from some known group G that can be expressed under the general formulation:

$$y = f(\mathsf{g} \circ x, x) \quad \text{for some} \quad \mathsf{g} = g(x) \in \mathsf{G} \qquad \text{(group-action learning)}$$

Above, $x \in \mathbb{R}^{d_{\text{in}}}$ and $y \in \mathbb{R}^{d_{\text{out}}}$ are input and output examples, $f, g$ are unknown functions, and $\circ$ denotes the application of a group action. As seen, the group element g can depend on the input data itself. More generally, the function $f$ may depend on more than one transformations of $x$ based on elements belonging to various groups of interest.

A simple instance of the group-action learning problem is presented in Figure 1. The example task is borrowed from ARC (Chollet, 2019) and entails learning to fill out the yellow patches in the leftmost image (input) so that the resulting image satisfies a 90° degree rotation symmetry. The learner is given only a small set of input-output pairs (the ARC tasks have 3.3 training examples on average) and the prior knowledge of discrete two-dimensional point groups, one of which is the cyclic group of 4-fold rotations $\mathsf{C}_4$. Though the task is challenging for a general neural network (due to the small number of samples), under the rotation prior it can be easily solved by the composition of a shallow

neural network, a group action, and a non-linear activation: 1) mapping yellow to zero, 2) rotating each image $x$ by some $g \in C_4$, and 3) taking a pixel-wise max.

It is important to stress that group-action learning is the exact antithesis of the typical group invariant and equivariant learning problems (Bronstein et al., 2021):

$$y = f(g \circ x) \quad \text{for } every \quad g \in G \qquad \qquad \text{(invariant learning)}$$

$$g \circ y = f(g \circ x) \quad \text{for } every \quad g \in G. \qquad \qquad \text{(equivariant learning)}$$

Intuitively speaking, whereas in group-action learning one aims to learn functions that involve specific (and data-dependent) transformations of our data by actions of the group, in in/equivariant learning the goal is to build models that are oblivious to such group actions in a systematic manner.

## 3  ATTENTION MASKS FOR CORE GEOMETRY PRIORS

This section prepares some theoretical grounding for LATFORMER, our approach to learn the transformations for lattice symmetry groups in the form of attention masks. The section defines attention masks and explains how the former can be leveraged to incorporate geometry priors when solving group action learning problems on sequences and images.

### 3.1  MODULATING ATTENTION WEIGHTS WITH SOFT MASKING

Consider the scaled dot-product attention mechanism as defined in Vaswani et al. (2017). In our formulation, we consider real-valued masks $M \in [0,1]^{n_Q \times n_K}$ that rescale attention weights:

$$A = \text{softmax}\left(\frac{QK^\top}{\sqrt{d}}\right) \odot M,$$

where $Q \in \mathbb{R}^{n_Q \times d}$ is the query parameter of the attention mechanism, $K \in \mathbb{R}^{n_K \times d}$ is the key, $d$ is the dimensionality of the model, $n_Q$ and $n_K$ are the sizes of the sets encoded by the query and key matrices respectively, and $\odot$ is the Hadamard product. Attention masks have been widely used to constrain the values of the attention weights and are usually binary masks applied before the $\text{softmax}$ activation (Vaswani et al., 2017; Sartran et al., 2022). However, as we aim to learn $M$, we apply the mask after the $\text{softmax}$ operation in order to avoid squashing the gradient. Therefore, we rescale the attention weights to sum to 1 when calculating the output $X$ of the attention mechanism:

$$\text{MaskedAttention}(Q, K, V; M) = \frac{A}{A \cdot \mathbf{1}_{n_K} \mathbf{1}_{n_K}^\top} V,$$

with $\mathbf{1}_n$ being a vector of ones of size $n$ and $V \in \mathbb{R}^{d \times n_K}$ being the value parameter of the attention mechanism. Though masking can also be applied in cross-attention, in the following we primarily focus on *self-attention*, where $Q = K = V = X$. For ease of notation, we write $\text{MaskedAttention}(X; M)$ whenever the query, key and value are the same matrix $X$.

### 3.2  EXISTENCE OF ATTENTION MASKS FOR LATTICE SYMMETRY ACTIONS

This section discusses group actions that can be represented by attention masks. To develop intuition, let us first consider the simple example of translation in a one-dimensional lattice. Supposing that $x = (x_1, \ldots, x_n)^\top$ is a vector of $n$ elements, we have:

$$\text{MaskedAttention}(x; M) = (x_n, x_1, \ldots, x_{n-1})^\top, \quad M = \begin{pmatrix} 0 & 0 & \cdots & 1 \\ 1 & 0 & \cdots & 0 \\ \vdots & \ddots & \ddots & \vdots \\ 0 & \cdots & 1 & 0 \end{pmatrix}.$$

Hence, when $M$ is the circulant permutation matrix shown above, we have that the mask shifts the input $x$ by one element to the right.

Beyond translation, it is natural to ask what kinds of group actions we can perform with attention masks on data with a more high-dimensional topological structure. The following theorem provides existence statements for data whose underlying topological space is a hypercubic lattice (such as sequences, images and higher-dimensional regular grids).

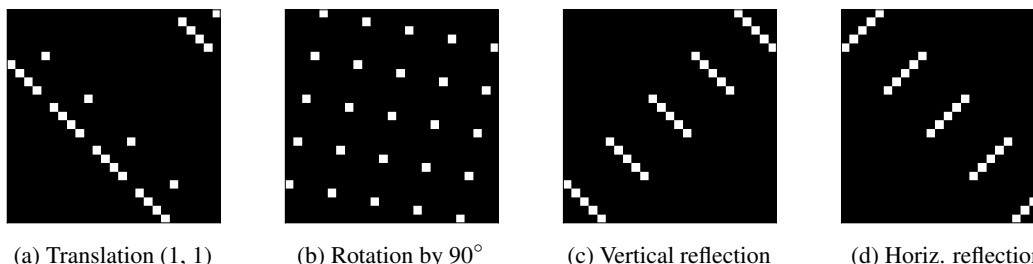

|     (a) Translation $(1, 1)$     |     (b) Rotation by $90°$     |     (c) Vertical reflection     |     (d) Horiz. reflection     |

Figure 2: Examples of attention masks implementing transformations in two dimensions, including: (a) translation by 1 pixel on both axes, (b) rotation by $90°$ counterclockwise, (c) vertical reflection and (d) horizontal reflection around the center. White represents value 1 and black 0.

| Transformation | Fourier shift | Size of $\mathbf{X}$ |
|---|---|---|
| **Identity** | $\boldsymbol{o}(\mathsf{g}_i)_k = 0$ | $n = l_1$ |
| **Translation** (by $\delta$) | $\boldsymbol{o}(\mathsf{g}_i)_k = -\delta$ | $n = l_1$ |
| **Reflection** | $\boldsymbol{o}(\mathsf{g}_i)_1 = (n-1), \quad \boldsymbol{o}(\mathsf{g})_k = \boldsymbol{o}(\mathsf{g})_{k-1} - 2$ | $n = l_1$ |
| **Rotation** ($90°$) | $\boldsymbol{o}(\mathsf{g}_i)_k = k \cdot (l_1 - 1) - \lfloor (k-1)/l_1 \rfloor$ | $n = l_1 \times l_2$ |
| **Upscaling** (by $h$) | $\boldsymbol{o}(\mathsf{g}_i)_k = (k - 1 \bmod h) + (h-1) \cdot \lfloor (k-1)/h \rfloor$ | $n = l_1$ |

Table 1: Fourier shifts for the transformations on the 1-dimensional and square lattices. We denote with $\boldsymbol{o}(\mathsf{g}_i)_k$ the $k$-th component of the vector $\boldsymbol{o}(\mathsf{g}_i) \in \mathbb{R}^n$, for $k = 1, \ldots, n$. As stated in Theorem 2, attention masks for higher-dimensional lattices can be obtained by the Kronecker product of primitive masks defined over the 1-dimensional and square lattices. Composition of actions is given by matrix multiplication of the masks.

**Theorem 1** (Existence). *Let $\mathsf{G}_m$ be the symmetry group of the $m$-dimensional hypercubic lattice, including translational symmetry, 4-fold rotational symmetry and vertical, horizontal and diagonal reflections. Let $\boldsymbol{X} \in \mathbb{R}^{n \times d}$ be a vectorized representation of an $m$-dimensional tensor $\mathbf{X} \in \mathbb{R}^{l_1 \times \cdots \times l_m}$, with $n = l_1 \cdot \ldots \cdot l_m$. For any group action $\mathsf{g} \in \mathsf{G}_m$, there exists an attention mask $\boldsymbol{M}_{\mathsf{g}} \in \{0, 1\}^{n \times n}$, such that:*

$$\mathrm{MaskedAttention}(\boldsymbol{X}; \boldsymbol{M}_{\mathsf{g}}) = \mathsf{g} \circ \boldsymbol{X}.$$

In other words, Theorem 1 states that any translation, rotation or reflection can be expressed in terms of an attention mask. Figure 2 shows some examples of masks corresponding to translation, rotation and reflection operations on square lattices.

In the following, we adopt the convention of writing $\boldsymbol{M}_{\mathsf{g}}$ to mean the mask that implements action $\mathsf{g}$. For more details and for a proof of Theorem 1, we refer the reader to Appendix D.

### 3.3 REPRESENTING ATTENTION MASKS FOR LATTICE TRANSFORMATIONS

To facilitate the learning of lattice symmetries, one needs to determine methods to parameterize the set of feasible group elements. Fortunately, as precised in the following theorem, the attention masks considered in Theorem 1 can be expressed conveniently under the same general formulation.

**Theorem 2** (Representation). *Let $\mathsf{G}_m$ be the symmetry group of the $m$-dimensional hypercubic lattice and $\mathsf{g} \in \mathsf{G}_m$ be an action on a tensor $\mathbf{X} \in \mathbb{R}^{l_1 \times \cdots \times l_m}$. Then, there exist some primitive attention masks $\boldsymbol{M}_{\mathsf{g}_i} \in \{0, 1\}^{n_i \times n_i}$ such that*

$$\boldsymbol{M}_{\mathsf{g}} = \bigotimes_i \boldsymbol{M}_{\mathsf{g}_i} \quad and \quad \mathcal{F}(\boldsymbol{M}_{\mathsf{g}_i}) = \mathcal{F}(\boldsymbol{I}_{n_i}) \exp(-\frac{2\pi j}{n_i} \, \boldsymbol{o}(\mathsf{g}_i) \, \boldsymbol{r}_{n_i}^{\top}),$$

*where $\boldsymbol{M}_{\mathsf{g}} \in \{0, 1\}^{n \times n}$ is an attention mask implementing $\mathsf{g}$, $\mathsf{g}_i \in \mathsf{G}_{m_i}$ for some $m_i \in \{1, 2\}$ is an action on the one-dimensional or square lattice, $\otimes$ is the Kronecker product, $\mathcal{F}$ is the Fourier transform applied column-wise, $\boldsymbol{I}_{n_i}$ is the $n_i \times n_i$ identity matrix, $j$ is the imaginary unit, $\boldsymbol{r}_{n_i} = (1, 2, \ldots, n_i)^{\top}$, and $\boldsymbol{o}(\mathsf{g}_i)$ is defined as in Table 1.*

To obtain an intuitive understanding of Theorem 2, it helps to revisit the example of translation by $\delta = 1$ of a sequence $\boldsymbol{x} \in \mathbb{R}^n$ on the 1-dimensional lattice ($m = 1$). Consulting Table 1, we find that $\boldsymbol{o}(\mathrm{g})$ is a vector containing $-1$ at every position and we know $\boldsymbol{M}_{\mathrm{g}}$ is the permutation circulant matrix of Section 3.2. Indeed, by the time-shifting property of the Fourier transform, $\boldsymbol{M}_{\mathrm{g}}$ can be obtained by shifting the rows of the identity by $-1$. In general, vector $\boldsymbol{o}(\mathrm{g})$ has a convenient intuitive interpretation as its $k$-th component represents the relative position (with respect to $k$) of the element that the $k$-th row of $\boldsymbol{X}$ attends to. For instance, in the one-dimensional example of translation by one element to the right, each element attends to the one immediately before. Hence, we have $\boldsymbol{o}(\mathrm{g})_k = -1$ for any $k = 1, \ldots, n$.

For higher-dimensional lattices, attention masks can be expressed as the Kronecker product of the attention masks for lower-dimensional cases. For instance, on the square lattice, a translation by 1 pixel on both dimensions is the Kronecker product of the two circulant matrices corresponding to a translation by 1 pixel on the one-dimensional lattice, as shown in Figure 2a. On more than one dimension, we can additionally define 4-fold rotations, still following the same formulation, with $\boldsymbol{o}(\mathrm{g}_i)$ defined as in Table 1.

Although strictly not a symmetry operation, *scaling transformations* of the lattice can also be defined in terms of attention masks under the same general formulation of Theorem 2, as reported in Table 1. Therefore, for completeness, we will consider scaling transformations as well in our experiments.

Notice that Theorem 2 allows us to derive a way to calculate the attention masks. In particular, we can express our attention masks as a convolution operation on the identity, as stated below.

**Corollary 1.** *Let* $\mathsf{G}_m$ *be the symmetry group of the* $m$*-dimensional hypercubic lattice and let* $\boldsymbol{M}_{\mathrm{g}} \in \mathbb{R}^{n \times n}$ *be an attention mask implementing action* $\mathrm{g} \in \mathsf{G}_m$*. Then:*

$$\boldsymbol{M}_{\mathrm{g}}[:, i] = \mathcal{F}^{-1}(\exp(-\frac{2\pi j}{n} \cdot \boldsymbol{o}(\mathrm{g}) \cdot \boldsymbol{r}_n^\top))[:, i] \star \boldsymbol{I}_n[:, i],$$

*where* $\star$ *denotes the convolution operation.*

In other words, we can represent any mask in our framework as a convolution of the identity matrix with predefined kernels. This motivates us to design a convolutional neural network that produces our attention masks by successive convolutions of the identity.

## 4 THE LATFORMER ARCHITECTURE

While in principle the problem of inferring group actions from input-output pairs can be solved via search over finite groups, in practice the size of the group for lattice symmetry actions makes this approach unfeasible[1]. Moreover, we are interested in learning unknown functions jointly with the transformation, which cannot be solved by searching on the space of group actions. Using a neural agent to search the space of possible actions would be a viable alternative, but this would make the problem non-differentiable and we would need to resort to reinforcement learning methods.

In this work, we aim to solve the problem in a *differentiable* way. Inspired by the observations above, we introduce LATFORMER, which incorporates the insights of Section 3 into a neural architecture that leverages our attention masks. We propose to use gated CNNs to parameterize the masks and we introduce an additional smoothing technique for easier optimization.

### 4.1 LATTICE MASK EXPERTS AS CONVOLUTIONAL NEURAL NETWORKS

Attention modules in neural networks usually include an attention mechanism with learnable linear transformations of the inputs[2] followed by a feed-forward network (FFN), as in the Transformer encoder layer (Vaswani et al., 2017).

To infuse core geometry priors in the attention module, we propose to modulate the attention weights with a mask generated by an additional layer, as shown in Figure 3a. We refer to this layer as *Lattice mask expert*, as it specializes towards specific transformations of the lattice. To understand the

---

[1]The size of the groups we consider grows with a polynomial of $n$ and exponentially with $m$.

[2]For simplicity, we omitted linear transformations in the definition of $\mathrm{MaskedAttention}$ in Section 3.1.

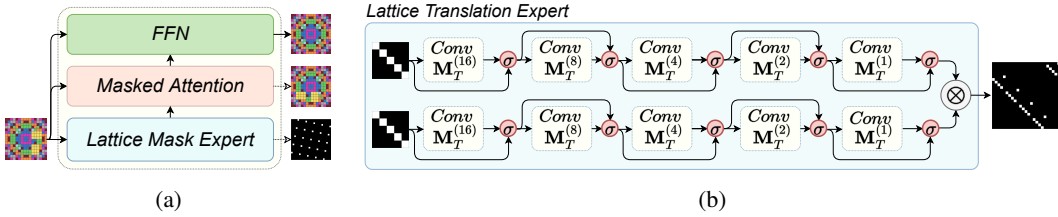

Figure 3: A LATFORMER layer (a) and an architecture for a *Lattice translation expert* (b). The LATFORMER layer (a) is a standard Transformer encoder layer augmented with a *Lattice mask expert* constrained to generate attention masks corresponding to a geometric transformation of the input. The *Lattice translation expert* (b) is a particular instance of a *Lattice mask expert* that produces translation masks. In the architecture above, every convolutional layer is meant to shift the input by a power of 2 and can be skipped by a gating function (denoted as $\sigma$).

purpose of this layer, it is useful to remember that, by the analysis conducted in Section 3, even if the attention and FFN layers are initialized to the identity function, the mask expert can generate attention masks that produce precise geometric transformations of the input.

By Corollary 1, we know that each group action on the lattice can be represented by a mask that is a convolution of the identity and we have an analytical expression to calculate the kernels of the convolution. We can leverage this notion to design CNNs that produce attention masks corresponding to specific group actions by following the general formulation:

$$\boldsymbol{M}_0 = \boldsymbol{I} \quad \text{and} \quad \boldsymbol{M}_{l+1} = \alpha_l \operatorname{Conv}(\boldsymbol{M}_l, \boldsymbol{K}_l) + (1 - \alpha_l)\boldsymbol{M}_l \ \text{ for } l = 0, \dots, L - 1,$$

where $\boldsymbol{M}_L$ is the predicted mask, $\alpha_l = \sigma_l(\boldsymbol{X}; \boldsymbol{\theta}) = \operatorname{FFN}_l(\boldsymbol{X}, \boldsymbol{\theta})$ is the output of a gating function, $\boldsymbol{\theta}$ is a learnable parameter, and $\boldsymbol{K}_l$ is the kernel of the $l$-th convolutional layer whose weights are determined based on Corollary 1 and Table 1.

As an example, Figure 3b shows an architecture that generates translation masks. Following Theorem 2, the expert computes the translations along the two dimensions separately and then aggregates the resulting masks doing the Kronecker product. Hence, a *Lattice translation expert* with $L$ convolutional layers for each dimension can generate any translation up to $\delta = 2^L - 1$ elements per dimension. At inference time, the values of the gates can be discretized, in such a way that the generated mask provably performs a meaningful group action.

Similarly to the expert in Figure 3b, we can define gated CNNs for transformations like rotation, reflection, and scaling. The product of experts (i.e., the combination of more actions) can be obtained by either chaining the experts or multiplying the attention masks generated by different experts. For more details, we refer the reader to Appendix A.

## 4.2 Mask smoothing for easier training

The framework described so far parameterizes discrete transformations of a lattice in a differentiable manner. Nevertheless, to improve the training of LATFORMER, we found it beneficial to also apply a smoothing operation on the attention masks w.r.t. the intrinsic metric of the group in question.

Our approach entails defining an adjacency relation between group elements and applying graph convolution with a heat kernel on the corresponding graph. This encourages the optimizer to favor weight updates that change the masks in a smooth manner w.r.t. the geodesic distance implied by the graph. Concretely, we define the neighbors of each element $\mathsf{g}_i$ on the lattice as those $\mathsf{g}_j = \mathsf{e} \circ \mathsf{g}_i$ reachable by an application of a primitive action e, such as translation by a single pixel in one dimension, rotation by $90°$, and vertical/horizontal reflection. The notion of neighborhood gives rise to a graph whose vertex set is the lattice group and that contains one edge for every pair of neighboring actions.

As before, it helps to consider different kinds of transformations separately. For instance, as shown in Figure 4, for 2D rotations the underlying graph is a cycle with 4 elements due to the underlying point group for 4-fold rotations being the cyclic group $\mathsf{C}_4$. Performing heat diffusion can be achieved by repeated neighborhood averaging over the cycle and yields a smoothed rotation

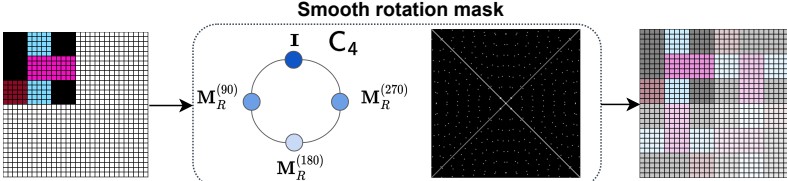

Figure 4: Rotational smoothing can be obtained by performing heat diffusion over a cyclic graph where each node corresponds to a rotation mask.

mask that performs all rotations at the same time (rightmost image in Figure 4). We can extend the same approach to all lattice transformations: for instance, in the case of translation, the underlying graph is a grid and the smoothing operation is akin to convolution with a Gaussian kernel.

To train LATFORMER with smoothed masks, we compute two predictions: one with the non-smooth mask predicted by the model and one with a smoothed version of the same mask. The final loss is the sum of two terms, one for the original prediction and one for the smoothed prediction. For smooth predictions, in all our experiments we utilize a mean squared error (MSE) loss, whereas for the original non-smooth predictions we use the cross-entropy or MSE loss depending on whether the output data are categorical or continuous.

## 5 EXPERIMENTS

To evaluate our method, we first developed a set of synthetic tasks in order to compare LATFORMER to attention modules and Transformers with respect to sample efficiency in learning basic geometric transformations. Then, we annotated the ARC tasks based on the knowledge priors they require, and we assessed the performance of our method on this challenging dataset. Finally, we experimented with the LARC (Acquaviva et al., 2021) dataset and compared our method to stronger baselines based on neural program synthesis. We report additional experimental results in Appendix B.3.

### 5.1 SAMPLE EFFICIENCY ON GEOMETRIC TRANSFORMATIONS

As a preliminary study, we probed the ability of LATFORMER to learn geometric transformations efficiently. To this end, we compared the performance of our model to a transformer (Vaswani et al., 2017) and an attention module (the same architecture as our approach, without the mask expert) on synthetic tasks with increasing number of examples. Inspired by ARC, we generated a set tasks where the model needs to infer a geometric transformation from input-output pairs. The input is a grid taken from the ARC tasks and the output is either a translation, rotation, reflection or scaling of the input. The specific transformation applied to the input grid defines the task and is consistent across all examples in the same task.

We evaluated the models based on the mean accuracy across tasks. Figure 5 shows the accuracy of our model compared to the baselines and to a version of LATFORMER without smoothing. The plots show that LATFORMER can generalize better and from fewer examples than transformers and attention modules both with absolute positional encodings and relative positional encodings (Shaw et al., 2018). Additionally, our results show that the smoothing operation described in Section 4.2 is helpful for larger groups. More details on this experiment are reported in Appendix B.1.

### 5.2 VISUAL REASONING ON ARC TASKS WITH GEOMETRIC KNOWLEDGE PRIORS

To assess the ability of our approach to learn efficiently on a more challenging use case, we focused on a subset of the ARC dataset (Chollet, 2019) requiring geometric priors for which our method could be a principled solution. To this end, we annotated the ARC tasks based on the knowledge priors they require, using the list of priors provided by Chollet (2019) as a reference. Appendix B.2 provides more details about the annotation of ARC and Figure 7a in the Appendix shows the knowledge priors that we considered and their distribution across the ARC tasks.

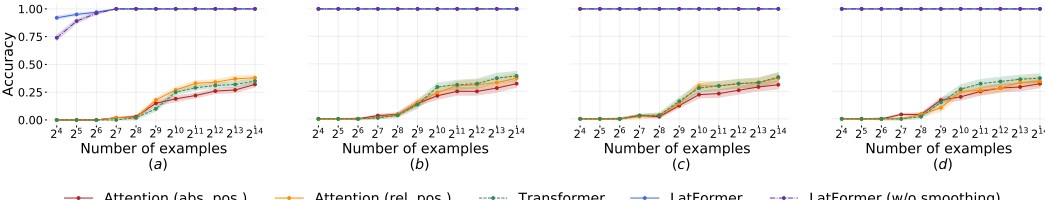

Figure 5: Sample efficiency of our method compared to the baselines on synthetic tasks on translation (a), rotation (b), reflection (c) and scaling (d). The $y$ axis denotes the mean accuracy across tasks belonging to the same category, whereas the error shade is the standard deviation.

| | Translate | Rotate + Translate | Reflect + Translate | Scale + Translate |
|---|---|---|---|---|
| **CNN** | 0.019 | 0.000 | 0.000 | 0.000 |
| **Attention (abs. pos.)** | 0.019 | 0.000 | 0.023 | 0.000 |
| **Attention (rel. pos.)** | 0.019 | 0.000 | 0.023 | 0.000 |
| **PixelCNN** | 0.019 | 0.000 | 0.000 | 0.000 |
| **Transformer** | 0.038 | 0.000 | 0.045 | 0.000 |
| **Transformer + data augmentation** | - | 0.200 | 0.184 | 0.091 |
| **LatFormer** | **0.365** | **0.800** | **0.591** | **0.545** |

Table 2: Performance on ARC tasks that involve lattice symmetry priors.

We assessed the performance of our model on the tasks that require only knowledge priors corresponding to the basic geometrical transformations that we addressed in this work, namely translation, rotation, reflection and scaling. Table 2 shows our results compared to neural baselines, including CNNs, attention with relative positional encodings (Shaw et al., 2018), PixelCNN (Gul et al., 2019), and Transformers (Vaswani et al., 2017). We additionally compared to a Transformer model that has access to precomputed transformations of the input (**Transformer + data augmentation**). Precomputing all group actions is only feasible for smaller groups (rotation, reflection and scaling). Though we restrict to only a subset of the tasks and there is definitely room for improvement even on these tasks, we reach considerably better performance than the baselines. Therefore, we believe our results advocate for the applicability of end-to-end differentiable models even on problems requiring sample-efficient abstract reasoning. To the extent of our knowledge, this is the first evidence of a neural network achieving this performance on ARC tasks.

## 5.3 Comparison with neural program synthesis on LARC

Recently, Acquaviva et al. (2021) introduced the *Language-complete Abstraction and Reasoning Corpus* (LARC), which provides natural language descriptions of 88% of the ARC tasks, generated by human participants who where asked to communicate to other humans a set of precise instructions to solve a task. Acquaviva et al. (2021) evaluated several models based on neural program synthesis on LARC. All models generate symbolic programs from a carefully designed domain-specific language (DSL). **LARC (IO)** is a model that has only access to input-output pairs, as our LATFORMER. **LARC (IO + NL)** has access to the natural language descriptions as well and uses a pre-trained T5 model (Raffel et al., 2020) to represent the text. **LARC (IO + NL pseudo)** uses *pseudo-annotation* to encourage the learning of compositional relationships between language and programs: during training, the model is given additional synthetic language-to-program pairs generated by annotating primitive examples in the DSL with linguistic comments.

In order to compare to the work of Acquaviva et al. (2021), we evaluated their models on the set of LARC tasks that correspond to ARC tasks in our subset requiring geometric knowledge priors. Additionally, following Acquaviva et al. (2021) we allowed LATFORMER to access the textual descriptions by using a pre-trained T5 model to generate a representation of the text. This embedding is provided as input both to the *Lattice Mask Expert* and the *FFN* layers of LATFORMER. We refer to this model as **LatFormer + NL**. Table 3 shows the results of our experiments on the LARC dataset. The program-synthesis methods require a training stage on a portion of the tasks. Therefore, the LATFORMER models where only evaluated on the same testing tasks of LARC, using the same train-test split of Acquaviva et al. (2021). Overall, our results shows that LATFORMER performs better than program synthesis on the subset of tasks requiring geometric priors, with no need for a

|                       | Translate | Rotate + Translate | Reflect + Translate | Scale + Translate |
|-----------------------|-----------|--------------------|---------------------|-------------------|
| **LARC (IO)**         | 0.17      | 0.00               | 0.42                | **1.00**          |
| **LARC (IO + NL)**    | 0.17      | 0.00               | 0.42                | **1.00**          |
| **LARC (IO + NL pseudo)** | 0.25  | 0.00               | 0.42                | **1.00**          |
| **LatFormer**         | **0.33**  | **1.00**           | 0.50                | **1.00**          |
| **LatFormer + NL**    | **0.33**  | **1.00**           | **0.58**            | **1.00**          |

Table 3: Comparison of LATFORMER with neural program synthesis methods with access to both input-output pairs and natural language descriptions on LARC

carefully designed DSL. This advantage comes to the expense of being restricted to tasks involving geometric priors, whereas program synthesis approaches can be used on a wider set of tasks. We also observe that the natural language descriptions marginally helped our model on one category of tasks. Our findings corroborate with Acquaviva et al. (2021) in this remark.

## 6 RELATED WORK

Our work was inspired by a previous investigation of self-attention layers which identified sufficient conditions such that they can perform convolution when equipped with relative positional encodings (Cordonnier et al., 2020; Andreoli, 2019). Rather than relying on relative encodings, we here show how soft-masking can be used to learn sample efficiently more general input transformations, such as rotation, reflection, and scaling.

To the extent of our knowledge, the group-action learning problem has not been explicitly and generally formulated in previous work. That being said, many previous works have focused on specific instances, such as learning to sort (Graves et al., 2014; Reed & De Freitas, 2015; Li et al., 2020) by selecting an element of the permutation group $S_n$, docking/folding by roto-translating objects according to an action in the special Euclidean group $\mathsf{SE}(3)$ (Sverrisson et al., 2022; Stärk et al., 2022; Jumper et al., 2021), and graph spectrum generation where the learned actions belong to the Stiefel manifold (Martinkus et al., 2022).

Our work is similar in spirit to recent efforts in neuro-symbolic visual reasoning (Johnson et al., 2017b;a; Goyal et al., 2017; Mao et al., 2019; Higgins et al., 2018). Many approaches based on attention mechanisms have been proposed in the past few years (Hudson & Manning, 2018; 2019). Our work differentiates from the former in that we aim to learn basic geometric reasoning in a sample-efficient way, rather than modeling relationships between high-level concepts.

Finally, some recent works came to our same conclusion on the advantages of using attention masks to incorporate prior knowledge in neural networks. As an example, Yan et al. (2020) focus on the task of learning subroutines (e.g., sorting algorithms) and use a CNN to generate an attention mask for a Transformer encoder. They show that learning the attention mask allows them to generalize to longer sequences than the ones provided at training time. Similarly, Sartran et al. (2022) used precomputed attention masks to incorporate syntactic compositional biases in language models.

## 7 CONCLUSION

This paper focused on how to help deep learning models to learn geometric transformations efficiently. Specifically, we proposed to incorporate lattice symmetry biases into attention mechanisms by modulating the attention weights using learned soft masks. We have shown that attention masks implementing the actions of the symmetry group of a hypercubic lattice exist, and we provided a way to represent these masks. This motivated us to introduce LATFORMER, a model that generates attention masks corresponding to lattice symmetry priors using a CNN. Our results on synthetic tasks show that our model can generalize better than the same attention modules without masking and Transformers. Moreover, the performance of our method on a subset of ARC provides the first evidence that deep learning can be used on this dataset, which is widely considered as an important challenge for AI research.

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

# A ADDITIONAL DETAILS ON THE MODEL

This section describes the LATFORMER architecture providing additional details that were not covered in Section 4.1. As mentioned in Section 4.1, it is possible to design convolutional neural networks that perform all considered transformations of the lattice. Figure 6 shows the architecture of the four expert models that generate *translation*, *rotation*, *reflection* and *scaling* masks.

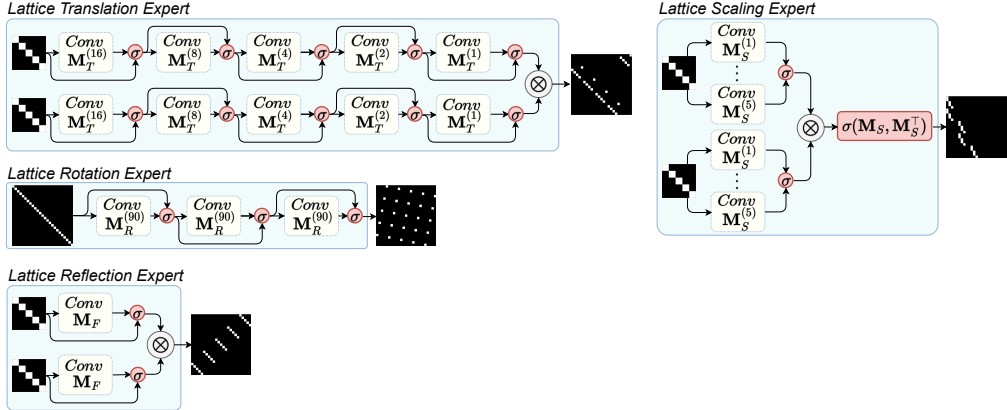

Figure 6: Model architecture of all the mask experts that we considered.

All models are CNNs applied to the identity matrix. In the figure, we use the following notation:

- $M_T^{(\delta)}$ denotes an attention mask implementing a translation by $\delta$ along one dimension;
- $M_R^{(90)}$ denotes an attention mask implementing a translation by $90°$;
- $M_F$ denotes an attention mask implementing a reflection along one dimension;
- $M_S^{(h)}$ denotes an attention mask implementing an upscaling by $h$ along one dimension.

Using Corollary 1, we can derive the kernels of the convolutional layers shown in Figure 6. These kernels are frozen at training time, the model only learns the gating function, denoted as $\sigma$ in the figure. Notice that all the models follow the same overall structure. However, for scaling, we also learn an additional gate, denoted as $\sigma(M_S, M_S^\top)$ in the Figure 6. This gate allows the model to transpose the mask and serves the purpose of implementing down-scaling operations (down-scaling is the transpose of up-scaling).

The composition of more actions can be obtained by combining different experts. This can be done either by chaining the experts or by matrix multiplication of the masks. In preliminary experiments, we did not notice any significant difference in performance between the two options and we rely on the latter in our implementation.

# B ADDITIONAL EXPERIMENTS AND DETAILS ON THE EXPERIMENTAL SETUP

This section provides additional details on the experimental setup of all our experiments, including further information on the generation of the synthetic tasks and the data annotation process for ARC.

## B.1 EXPERIMENTS ON SYNTHETIC DATA

We considered four categories of tasks, namely *translation*, *rotation*, *reflection* and *scaling*. Each task is defined in terms of input-output pairs, which are sampled from the set of all ARC grids and padded to the size of $30 \times 30$ cells. To each input grid, a synthetic transformation is applied in order to obtain the corresponding output grid. For each task in each category, we generated 2048 training pairs and 100 test pairs.

For translation tasks, we have a total of 900 possible translations in a $30 \times 30$ grid. However, generating data and training models on 900 tasks is computationally expensive, so we randomly

sampled 5 translations in the interval $[1, 29] \times [1, 29]$, obtaining a total of 100 translation tasks. Rotation tasks include all 4-fold rotations except the identity. Similarly, reflection tasks involve horizontal, vertical and diagonal reflections. Scaling tasks include all possible up/down scaling transformations of the input grid by factors of $[2, 5] \times [2, 5]$ for a total of 32 scaling tasks.

The models are evaluated based on the mean accuracy on each category. For each task we compute the accuracy on the test set based on how many of the predicted images match exactly the ground truth.

## B.2 EXPERIMENTS ON ARC

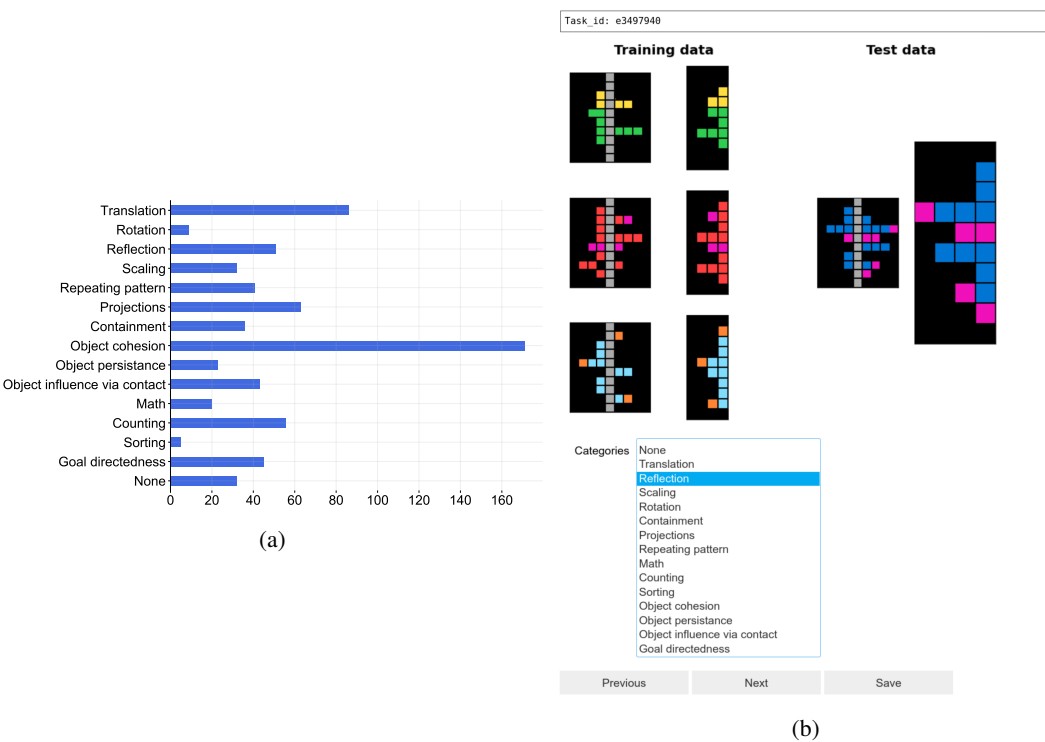

Figure 7: Distribution of the considered core knowledge priors across the ARC tasks (a) and user interface built to annotate the dataset (b).

In order to experiment with ARC, we first performed an annotation of the dataset to identify the underlying knowledge priors for each task. To this end, we built a user interface where the annotator could browse the tasks and label them by selecting any combination of the available knowledge priors. Figure 7b shows the user interface provided to the annotator, whereas Figure 7a shows the distribution of knowledge priors across the ARC tasks. Most tasks follow in more than one of the categories represented in Figure 7a.

ARC can be regarded as a meta-learning benchmark, as it provides a set of training tasks and a set of unseen tasks to evaluate the performance of the model learned on the meta-training data. It is important to emphasize that we do not target this use case, as we instead use the same setup as in the synthetic data and learn each task from scratch using only its training set.

Though simple and elegant, the supervised-learning formulation prevents our models from reusing knowledge that can be shared between different tasks. In order to mitigate this issue, we rely on a data-augmentation strategy. At training time, for each model and every iteration, we augment each grid 10 times by mapping each color to a different color (using the same mapping across training examples). The rationale behind this data-augmentation strategy is that *(1)* we assume that for tasks involving only geometric knowledge priors to be not affected by color mapping and *(2)* all models (including LATFORMER) need to learn a function from $d$-dimensional color representations to categorical variables, hence it is beneficial if all colors are represented in the training set.

| | **Aerial data** | | | **Cytological data** | | |
|---|---|---|---|---|---|---|
| | $\alpha$-**AMD** | **SIFT** | **LatFormer** | $\alpha$-**AMD** | **SIFT** | **LatFormer** |
| **CycleGAN** $(A \rightarrow B)$ | $5.3 \pm 3.1$ | $67.2 \pm 16.8$ | $\mathbf{68.3 \pm 4.5}$ | $\mathbf{74.2 \pm 3.8}$ | $30.2 \pm 4.2$ | $68.3 \pm 2.2$ |
| **CycleGAN** $(B \rightarrow A)$ | $65.7 \pm 6.7$ | $84.0 \pm 2.5$ | $\mathbf{86.1 \pm 3.1}$ | $21.3 \pm 1.8$ | $18.2 \pm 3.5$ | $\mathbf{24.2 \pm 3.3}$ |
| **DRIT++** $(A \rightarrow B)$ | $35.3 \pm 2.4$ | $38.1 \pm 8.1$ | $\mathbf{38.2 \pm 5.9}$ | $50.4 \pm 12.1$ | $24.2 \pm 2.7$ | $\mathbf{62.7 \pm 10.2}$ |
| **DRIT++** $(B \rightarrow A)$ | $20.2 \pm 2.1$ | $38.3 \pm 4.5$ | $\mathbf{43.2 \pm 4.1}$ | $\mathbf{30.1 \pm 4.5}$ | $5.2 \pm 3.1$ | $15.6 \pm 3.5$ |
| **pixel2pixel** $(A \rightarrow B)$ | $84.2 \pm 4.0$ | $\mathbf{98.7 \pm 0.4}$ | $89.3 \pm 2.2$ | $53.2 \pm 6.9$ | $9.5 \pm 1.0$ | $\mathbf{61.2 \pm 5.5}$ |
| **pixel2pixel** $(B \rightarrow A)$ | $68.2 \pm 7.5$ | $87.5 \pm 4.03$ | $\mathbf{89.7 \pm 3.3}$ | $0.2 \pm 0.1$ | $4.0 \pm 1.0$ | $\mathbf{4.2 \pm 1.1}$ |
| **StarGAN** $(A \rightarrow B)$ | $63.1 \pm 7.8$ | $7.4 \pm 2.7$ | $\mathbf{72.2 \pm 6.3}$ | $\mathbf{60.2 \pm 12.2}$ | $12.2 \pm 2.0$ | $59.5 \pm 5.9$ |
| **StarGAN** $(B \rightarrow A)$ | $52.1 \pm 4.0$ | $7.9 \pm 1.3$ | $\mathbf{53.3 \pm 4.0}$ | $\mathbf{20.8 \pm 3.9}$ | $4.1 \pm 0.9$ | $13.4 \pm 3.1$ |
| **CoMIR** | $94.2 \pm 5.7$ | $\mathbf{100.0 \pm 0.0}$ | $90.2 \pm 3.3$ | $76.2 \pm 12.1$ | $74.1 \pm 6.3$ | $\mathbf{78.1 \pm 3.4}$ |

Table 4: Results of the experiment on image registration. The rows represent different models trained to translate images from modality A to B $(A \rightarrow B)$ or viceversa $(B \rightarrow A)$.

All models are evaluated based on the ratio of solved tasks and a task is considered solved if the model can predict the correct output grid for *all* examples in the test set.

### B.3    ADDITIONAL EXPERIMENTS ON IMAGE REGISTRATION

As an additional experiment, to assess the applicability of our LATFORMER on natural images, we performed experiments on multimodal *image registration*, namely the problem of spatially aligning images from different modalities. Image registration is a well-studied problem in computer vision and we do not aim to establish state-of-the-art performance. The main purpose of this experiment is giving a hint on the applicability of our method to natural images beyond ARC. We refer the reader to SuperGlue (Sarlin et al., 2020) and COTR (Jiang et al., 2021) to have a sense of approaches specifically designed for this task.

Popular approaches to multimodal image registrations work in two stages: first, they learn a model that converts one modality into the other (or to transfer both modalities in the same representation as proposed by Pielawski et al. (2020)), then they align the two images using traditional techniques. We follow the experimental setup of Lu et al. (2021) and experiment with two datasets, one containing aerial views of a urban neighborhood and one containing cytological images. The images we employ are views of the same scene, but they are taken with different modalities and they are translated with respect to one another. We use the code of the authors to generate data involving only translations. Lu et al. (2021) additionally consider small rotations, but these transformations are not actions in the symmetry group of a lattice, so we are not interested in resolving them.

We employ several state-of-the-art methods for modality translation and we compare our method to $\alpha$-AMD (Lindblad & Sladoje, 2014) and SIFT (Lowe, 1999) based on the success rate metric defined by Lu et al. (2021). A registration is considered successful if the relative registration error (i.e., the residual distance between the reference patch and the transformed patch after registration normalized by the height and width of the patch) is below $2\%$. Table 4 reports our results on the image registration tasks and shows that our approach performs well on both datasets coupled with different methods for modality translation. We use the same models of Lu et al. (2021) for the modality translation task. Then, in order to solve the image registration task with LATFORMER, we divide each image into $30 \times 30$ patches and we run our model to predict the translation from one patch in an image to its counterpart in the corresponding image.

## C    LIMITATIONS AND FUTURE WORK

Although we believe our results are interesting and promising for learning group actions with neural networks, we would like to point out some limitations of our approach.

First, our method is limited to actions on the symmetry group of the hypercubic lattice and it is not immediately extendable to other groups. For instance, though permutation matrices are still convolutions of the identity and they can be generated by a CNN, providing an architecture with predefined kernels that can compute any permutation matrix is not feasible. Second, the model is

hard to fine-tune: we noticed that once the gates of the CNN have been trained, it is hard for the model to adapt to different actions.

We believe that both limitations can be addressed by still keeping the same overall idea of modulating attention weights using soft attention masks, possibly with a different parametrization of the masks. Future work will focus on this research direction and on extending our work to cover a wider set of the ARC tasks.

## D DEFERRED PROOFS

We prove both Theorem 1 and 2 by induction on the dimensionality of the hypercubic lattice $m$.

### D.1 BASE CASE FOR THEOREMS 1 AND 2

First, it is useful to notice that whenever $\boldsymbol{M} \in \{0,1\}^{n \times n}$ has exactly a single 1 per row, in other words $\boldsymbol{M} \cdot \boldsymbol{1}_n = \boldsymbol{1}_n$, then, for any $\boldsymbol{X} \in \mathbb{R}^{n \times d}$

$$
\begin{aligned}
\mathrm{MaskedAttention}(\boldsymbol{X}; \boldsymbol{M}) &= \frac{\boldsymbol{A}}{\boldsymbol{A} \cdot \boldsymbol{1}_n \boldsymbol{1}_n^\top} \boldsymbol{X} \\
&= \frac{\mathrm{softmax}\left(\frac{\boldsymbol{X}\boldsymbol{X}^\top}{\sqrt{d}}\right) \odot \boldsymbol{M}}{\mathrm{softmax}\left(\frac{\boldsymbol{X}\boldsymbol{X}^\top}{\sqrt{d}}\right) \odot \boldsymbol{M} \cdot \boldsymbol{1}_n \boldsymbol{1}_n^\top} \boldsymbol{X} \\
&= \boldsymbol{M} \cdot \boldsymbol{X}.
\end{aligned}
$$

In order to prove the theorem, we need to show that, for any action $\mathsf{g} \in \mathsf{G}_1$, including translations, reflections and rotations, there exists a mask $\boldsymbol{M}_\mathsf{g}$ such that:

$$
\mathrm{MaskedAttention}(\boldsymbol{X}; \boldsymbol{M}_\mathsf{g}) = \mathsf{g} \circ \boldsymbol{X}.
$$

Let us consider different families of actions separately.

**Translation.** As mentioned in Section 3.2, in the 1-dimensional case, a translation by one element to the right for a vector $\boldsymbol{x} = (x_1, x_2, \ldots, x_n)^\top$ is given by the circulant permutation matrix:

$$
\boldsymbol{M} = \boldsymbol{M}_T^{(1)} = \begin{pmatrix} 0 & 0 & 0 & \cdots & 1 \\ 1 & 0 & 0 & \cdots & 0 \\ 0 & 1 & 0 & \cdots & 0 \\ \vdots & \ddots & \ddots & \ddots & \vdots \\ 0 & \cdots & 0 & 1 & 0 \end{pmatrix}.
$$

This holds because $\boldsymbol{M}_T^{(1)} \cdot \boldsymbol{1}_n = \boldsymbol{1}_n$, so:

$$
\mathrm{MaskedAttention}(\boldsymbol{x}; \boldsymbol{M}_T^{(1)}) = \boldsymbol{M}_T^{(1)} \cdot \boldsymbol{x} = (x_n, x_1, x_2, \ldots, x_{n-1})^\top.
$$

In general, a translation by $\delta$ elements is given by the circulant matrix $\boldsymbol{M}_T^{(\delta)} = (\boldsymbol{M}_T^{(1)})^\delta$. Therefore, masks implementing translation operations exist in the 1-dimensional case and they are circulant permutation matrices. This is enough for a base case for Theorem 1.

For Theorem 2, simply notice that:

$$
\boldsymbol{M}_T^{(\delta)} = \mathcal{F}^{-1}\left(\mathcal{F}(\boldsymbol{I}_n) \exp\left(-\frac{2\pi j}{n} \boldsymbol{o}_T^{(\delta)} \boldsymbol{r}_n^\top\right)\right) \quad \text{where} \quad \boldsymbol{o}_T^{(\delta)} = \begin{pmatrix} -\delta \\ -\delta \\ \vdots \\ -\delta \end{pmatrix}.
$$

This comes directly from the time-shifting property of the Fourier transform.

**Reflection.** In the 1-dimensional case, the reflection of a vector $\boldsymbol{x} = (x_0, x_1, \ldots, x_n)^\top$ is:

$$\text{MaskedAttention}(\boldsymbol{x}; \boldsymbol{M}_F) = \boldsymbol{M}_F \cdot \boldsymbol{x} = (x_n, x_{n-1}, \ldots, x_2, x_1)^\top$$

with

$$\boldsymbol{M}_F = \begin{pmatrix} 0 & \cdots & 0 & 0 & 1 \\ 0 & \cdots & 0 & 1 & 0 \\ 0 & \cdots & 1 & 0 & 0 \\ \vdots & \cdots & \vdots & \vdots & \vdots \\ 1 & \cdots & 0 & 0 & 0 \end{pmatrix}.$$

The attention mask $\boldsymbol{M}_F$ can be obtained by shifting the rows of the identity matrix by:

$$\boldsymbol{o}_F = \begin{pmatrix} n-1 \\ n-3 \\ n-5 \\ \vdots \\ 1 \end{pmatrix}.$$

Therefore, by the time-shifting property of the Fourier transform we have:

$$\boldsymbol{M}_F = \mathcal{F}^{-1}\big(\mathcal{F}(\boldsymbol{I}_n) \exp(-\frac{2\pi j}{n} \boldsymbol{o}_F \, \boldsymbol{r}_n^\top)\big).$$

**Rotation.** Rotation (4-fold) is not defined in one dimension, so for a base case we need to consider the square lattice. Let $\boldsymbol{X} \in \mathbb{R}^{l_1 \cdot l_2}$ be a vectorized representaiton of a $n = l_1 \times l_2$ dimensional matrix. We need to define a vector $\boldsymbol{o}_R \in \mathbb{R}^n$ such that:

$$\boldsymbol{M}_R^{(90)} = \mathcal{F}^{-1}\big(\mathcal{F}(\boldsymbol{I}_n) \exp(-\frac{2\pi j}{n} \boldsymbol{o}_R \, \boldsymbol{r}_n^\top)\big)$$

is a rotation mask. Since rotation is a permutation of the identity, we know the vector exists. As $\boldsymbol{X}$ is vectorized, the $\boldsymbol{o}_R^{(90)}$ needs to take into account the size of the first dimension $l_1$. For example, in order to perform a rotation on a vectorized representation, we need to map the first element of $\boldsymbol{X}$ to the position $(l_1 - 1)$. The reader can check that the vector given by

$$(\boldsymbol{o}_R^{(90)})_k = k \cdot (l_1 - 1) - \lfloor (k-1)/l_1 \rfloor$$

satisfies the equation above.

**Scaling.** Although scaling is not a group action of the symmetry group of the lattice, we pointed out that it still can be defined within the same general formulation as the other transformations. We can take the 1-dimensional lattice as a base case and consider a vector $\boldsymbol{x} = (x_0, x_1, \ldots, x_n)^\top$. Let $h \in \mathbb{N}$ be a parameter specifying the filter size of the scaling operation. As an example, for $h = 2$ we have:

$$\text{MaskedAttention}(\boldsymbol{x}; \boldsymbol{M}_S^{(h)}) = \boldsymbol{M}_S^{(h)} \cdot \boldsymbol{x} = (x_1, x_1, x_2, x_2, \ldots, x_{\lfloor n/2 \rfloor})^\top,$$

where:

$$\boldsymbol{M}_S^{(h)} = \begin{pmatrix} 1 & 0 & \cdots & 0 & 0 \\ 1 & 0 & \cdots & 0 & 0 \\ 0 & 1 & \cdots & 0 & 0 \\ 0 & 1 & \cdots & 0 & 0 \\ \vdots & \vdots & \cdots & \vdots & \vdots \\ 0 & 0 & \cdots & 0 & 0 \end{pmatrix}.$$

This kind of matrix can also be obtained by shifting the rows of the identity as follows:

$$\boldsymbol{M}_S^{(h)} = \mathcal{F}^{-1}\big(\mathcal{F}(\boldsymbol{I}_n) \exp(-\frac{2\pi j}{n} \boldsymbol{o}_S^{(h)} \, \boldsymbol{r}_n^\top)\big),$$

where $\boldsymbol{o}_S^{(h)} = (k - 1 \bmod h) + (h - 1) \cdot \lfloor (k-1)/h \rfloor$.

## D.2 INDUCTIVE STEP FOR THEOREMS 1 AND 2

Suppose that $\boldsymbol{M}_{\mathsf{g}_1} \in \{0,1\}^{n_1 \times n_1}$ and $\boldsymbol{M}_{\mathsf{g}_2} \in \{0,1\}^{n_2 \times n_2}$ are attention masks implementing actions $\mathsf{g}_1 \in \mathsf{G}_{m_1}$ and $\mathsf{g}_2 \in \mathsf{G}_{m_2}$ on some tensors $\mathsf{X}_1 \in \mathbb{R}^{l_1 \times \cdots \times l_{m_1}}$ and $\mathsf{X}_2 \in \mathbb{R}^{l'_1 \times \cdots \times l'_{m_2}}$, with $n_1 = l_1 \cdot \ldots \cdot l_{m_1}$ and $n_2 = l'_1 \cdot \ldots \cdot l'_{m_2}$. Consider a tensor $\mathsf{X} \in \mathbb{R}^{l_1 \times \cdots \times l_{m_1} \times l'_1 \times \cdots \times l'_{m_2}}$ and its vectorization $\boldsymbol{X} \in \mathbb{R}^n$ with $n = n_1 n_2$.

We have:

$$
\begin{aligned}
\mathrm{MaskedAttention}(\boldsymbol{X}; \boldsymbol{M}_{\mathsf{g}_1} \otimes \boldsymbol{M}_{\mathsf{g}_2}) &= \\
&= (\boldsymbol{M}_{\mathsf{g}_1} \otimes \boldsymbol{M}_{\mathsf{g}_2})\,\boldsymbol{X} \\
&= (\boldsymbol{M}_{\mathsf{g}_1} \otimes \boldsymbol{I}_{n_2})(\boldsymbol{I}_{n_1} \otimes \boldsymbol{M}_{\mathsf{g}_2})\boldsymbol{X} \\
&= \mathrm{MaskedAttention}(\mathrm{MaskedAttention}(\boldsymbol{X}; \boldsymbol{I}_{n_1} \otimes \boldsymbol{M}_{\mathsf{g}_2}); \boldsymbol{M}_{\mathsf{g}_1} \otimes \boldsymbol{I}_{n_2}).
\end{aligned}
$$

Now notice that:

$$
\begin{aligned}
\mathrm{MaskedAttention}(\boldsymbol{X}; \boldsymbol{I}_{n_1} \otimes \boldsymbol{M}_{\mathsf{g}_2}) &= (\boldsymbol{I}_{n_1} \otimes \boldsymbol{M}_{\mathsf{g}_2})\,\boldsymbol{X} \\
&= \mathrm{vec}(\boldsymbol{M}_{\mathsf{g}_2}\,\mathrm{vec}^{-1}(\boldsymbol{X})\,\boldsymbol{I}_{n_1}) \\
&= \mathrm{vec}(\boldsymbol{M}_{\mathsf{g}_2}\,\mathrm{vec}^{-1}(\boldsymbol{X})),
\end{aligned}
$$

and similarly

$$
\begin{aligned}
\mathrm{MaskedAttention}(\boldsymbol{X}; \boldsymbol{M}_{\mathsf{g}_1} \otimes \boldsymbol{I}_{n_2}) &= (\boldsymbol{M}_{\mathsf{g}_1} \otimes \boldsymbol{I}_{n_2})\,\boldsymbol{X} \\
&= \mathrm{vec}(\boldsymbol{I}_{n_2}\,\mathrm{vec}^{-1}(\boldsymbol{X})\,\boldsymbol{M}_{\mathsf{g}_1}^{\top}) \\
&= \mathrm{vec}((\boldsymbol{M}_{\mathsf{g}_1}\,\mathrm{vec}^{-1}(\boldsymbol{X})^{\top})^{\top}).
\end{aligned}
$$

Therefore, we conclude that performing masked attention with the mask $\boldsymbol{M}_{\mathsf{g}_1} \otimes \boldsymbol{M}_{\mathsf{g}_2}$ on $\mathsf{X}$ is equivalent to applying $\mathsf{g}_1$ on the first $m_1$ dimensions and $\mathsf{g}_2$ on the last $m_2$ dimensions of $\mathsf{X}$. This provides a way for building attention masks for higher-dimensional lattices using the primitive masks defined in Section D.1, proving both Theorem 1 and 2.

## D.3 PROOF OF COROLLARY 1

The proof of Corollary 1 follows immediately from Theorem 2 and from the property of the Fourier transform according to which multiplying in the Fourier domain implements a convolution in the original domain.

