# OpenReview forum: "Infusing Lattice Symmetry Priors in Neural Networks Using Soft Attention Masks"
_ICLR.cc/2023/Conference — Submitted to ICLR 2023_

### Official Review · Reviewer_9teR · 2022-10-25

**Confidence:** 4
**Correctness:** 3
**Technical Novelty And Significance:** 2
**Empirical Novelty And Significance:** 2
**Recommendation:** 6

**Clarity, Quality, Novelty And Reproducibility:**

The paper reads well. The proposed method makes sense. Code is not submitted and hence reproducibility is not guaranteed.

**Strength And Weaknesses:**

Strengths:
1. The paper is evaluated on a number of tasks.
2. Results show that Latformer can learn geometric transformations with fewer training data.

Weaknesses:
1. Table 3 only compares the proposed method to alpha-AMD and SIFT which are old methods. Since this is an image registration task, the proposed method should be compared to more recent methods, e.g., COTR: Correspondence transformer for matching across images. ICCV 2021.
2. The paper should also discuss failure modes, limitations of the proposed method and future work.

**Summary Of The Paper:**

This paper proposes LAtFormer, which incorporates lattice geometry and topology priors in attention masks. The proposed architecture is a modification of the standard attention mechanism where attention weights are scaled using soft attention masks generated by a convolution neural net. Experiments on ARC and synthetic visual reasoning tasks show that LatFormer requires 2 orders of magnitude fewer data compared to standard attention and transformers.

**Summary Of The Review:**

Please see the comments in the boxes above.

---

> ### Author Response · Authors · 2022-11-19
> **Reply to Reviewer 9teR**
>
> Thanks for the time spent on our manuscript and for the feedback. We improved the paper by making the scope of our work more clear and by including additional results. We kindly ask the reviewer to have a look at the updated version of the paper and the general reply. We answer your further questions below.
>
> > Table 3 only compares the proposed method to alpha-AMD and SIFT which are old methods. Since this is an image registration task, the proposed method should be compared to more recent methods, e.g., COTR: Correspondence transformer for matching across images. ICCV 2021.
>
> As explained in our general reply, the main contribution of our work is not a method for image registration and we do not claim to be state of the art on this task. We addressed this comment in two ways. First, we moved the experiment on image registration to the appendix, we better explained the purpose of the experiment and added a reference to COTR (see Appendix B.3). Then, we improved the paper to make the scope more clear and we added more experimental results on ARC and LARC in order to compare our method against strong baselines based on neural program synthesis. Please see our general reply and the updated version of the paper for more details.
>
> >The paper should also discuss failure modes, limitations of the proposed method and future work.
>
> Thanks for this comment, we added a discussion of the main limitations and future directions in the Appendix (please see Appendix C).

---

> > ### Comment · Reviewer_9teR · 2022-11-22
> > **Post rebuttal comments**
> >
> > Most of my concerns have been addressed. Therefore, I'd like to raise my rating to 6.

---

> > > ### Author Response · Authors · 2022-11-28
> > > **Thanks**
> > >
> > > Thanks for your positive feedback and for increasing your rating!

---

### Official Review · Reviewer_YP2T · 2022-10-25

**Confidence:** 4
**Correctness:** 4
**Technical Novelty And Significance:** 3
**Empirical Novelty And Significance:** 3
**Recommendation:** 6

**Clarity, Quality, Novelty And Reproducibility:**

The paper is clearly written and easy to follow. The code is not provided. The experiments on ARC tasks require extra annotations which is not public available.

**Strength And Weaknesses:**

Strength:

+ The proposed method presents an interesting way to infuse inductive biases and knowledge priors in deep neural networks, which is an important open problem for next step's developments of AI and neural networks.

+ The idea of using a convolutional neural network to generate soft attention masks to scaled attention weights is reasonable.

+ Experimental results on synthetic tasks and ARC tasks show that the proposed LatFormer can generalize better and from fewer
examples than transformers and unmasked attention modules. Experiments on image registration task further validates the applicability of LatFormer on natural images.


Concerns:

- LatFormer shows potential of inductive biases and knowledge priors. I'm wondering if LatFormer also performs better than learning transformations invariant representations by applying augmentations? Do we have such studies on the current experimental settings/tasks?

-  For image registration experiments in Section 5.3, how about reuslts of using the baseline Transformer?

- How about the application of the proposed LatFormer to other reasoning tasks, e.g. Physical Reasoning in [1]

[1] Physical Reasoning Using Dynamics-Aware Models, ICML 2021

**Summary Of The Paper:**

This paper focuses on helping deep learning models learn fundamental geometric transformations efficiently. Specifically, LatFormer is proposed to incorporate lattice symmetry biases into attention mechanisms by modulating the attention weights using learned soft masks. Experiments are conducted on both synthetic tasks and image registration tasks to show that the proposed model can generalize better than the same attention modules without masking and Transformers. Besides, LatFormer also shows potential on ARC tasks that incorporates geometric priors.

**Summary Of The Review:**

This is an interesting paper about learning geometric transformations with deep neural networks, with experimental evidences from both synthetic tasks, visual reasoning tasks as well as applications on natural images.

---

> ### Author Response · Authors · 2022-11-19
> **Reply to Reviewer YP2T**
>
> Thanks for the time spent on our manuscript and for the overall positive feedback. We believe that we considerably improved the initial version of the paper and we kindly ask you to have a look at the updated version and the general reply. We provide answers to your specific comments below.
>
> > LatFormer shows potential of inductive biases and knowledge priors. I'm wondering if LatFormer also performs better than learning transformations invariant representations by applying augmentations? Do we have such studies on the current experimental settings/tasks?
>
> Thanks for your insightful suggestion. We performed this additional experiment and we reported the result in Table 2. We used a transformer model with access to all precomputed transformations of the input on ARC (so that the model needs to learn which transformation to copy). This model is referred to as **Transformer + data augmentation** in Table 2. Notice that this is only feasible for small groups (like rotations, reflections and scaling by some small factor) while it is unfeasible for large groups (like translation). For convenience, we report the results below. The experiment shows that our model can learn more efficiently than a Transformer with access to precomputed transformations of the input.
>
> |                                     | **Translate** | **Rotate + Translate** | **Reflect + Translate** | **Scale + Translate** |
> |-------------------------------------|---------------|------------------------|-------------------------|-----------------------|
> | **Transformer**                     | 0.038         | 0.000                  | 0.045                   | 0.000                 |
> | **Transformer + data augmentation** | -             | 0.200                  | 0.184                   | 0.091                 |
> | **LatFormer**                       | **0.365**     | **0.800**                | **0.591**               | **0.545**             |
>
>
> > For image registration experiments in Section 5.3, how about results of using the baseline Transformer?
>
> For the experiment on image registration, we used the same experimental setup of Lu et al. (2021), with the same datasets and the same models. We performed an additional experiment with a vanilla transformer encoder as suggested, and then we recovered the transformation performed by the model from the attention matrix. The results are given in the table below and show that a vanilla transformer is not able to learn the task from just the few patches of an input and output image.
>
> |                           | **Aerial data** | **Cytological data** |
> |---------------------------|-----------------|----------------------|
> | **CycleGAN (A -> B)**     | 0.8             | 4.2                  |
> | **CycleGAN (B -> A)**     | 5.1             | 3.2                  |
> | **DRIT++ (A -> B)**       | 1.7             | 3.1                  |
> | **DRIT++ (B -> A)**       | 2.2             | 3.5                  |
> | **pixel2pixel (A -> B)**  | 2.1             | 2.2                  |
> | **$pixel2pixel (B -> A)** | 3.9             | 0.0                  |
> | **StarGAN (A -> B)**      | 4.3             | 1.7                  |
> | **StarGAN (B -> A)**      | 1.9             | 0.5                  |
> | **CoMIR**                 | 6.1             | 5.3                  |
>
> > How about the application of the proposed LatFormer to other reasoning tasks, e.g. Physical Reasoning
>
> We thank the reviewer for this suggestion. We did not perform additional experiments for this task because we believe that adapting the model would be non-trivial, but we think this could be an interesting avenue for future research.

---

> > ### Author Response · Authors · 2022-11-28
> > **Reminder**
> >
> > Thanks again for the time spent on our manuscript and for your feedback. We clarified the scope of the paper and included additional experiments. Overall, we believe that the current version of the paper should address your initial concerns and we are looking forward to your feedback on our reply. Please let us know if you have any additional comments.

---

> > ### Comment · Reviewer_YP2T · 2022-12-13
> > **Thanks for your reply**
> >
> > I appreciate responses and experimental results from authors, which helps to solve some of my concerns. Therefore, I'd like to keep score 6.

---

### Official Review · Reviewer_ch2n · 2022-10-25

**Confidence:** 3
**Correctness:** 3
**Technical Novelty And Significance:** 3
**Empirical Novelty And Significance:** 3
**Recommendation:** 6

**Clarity, Quality, Novelty And Reproducibility:**

The paper is clear and the ideas appear to be novel. It seems reasonably easy to reproduce. Many experimental details are provided in the appendix.

**Strength And Weaknesses:**

Strength
* The idea of introducing structural priors in the form of attention masks for geometric transformation tasks is intuitive and well-motivated
* The theoretical results show that such priors can expressed as a convolution on an identity matrix, which motivates the design of LatFormer
* Empirical results show remarkable sample efficiency gains over vanilla transformers in ARC tasks and outperform baselines on real image registration

Weakness
* It’d be valuable to show more experiments on real data. How about showing the results on standard image registration/correspondence learning benchmarks?
* In Table 3, are there other baselines to compare with? I’d imagine a vanilla transformer would be a reasonable one.
* In Figure 5, it seems that the transformer is barely learning the task. I’m curious how many samples would it take for a transformer to learn this task perfectly (how far should we extend the x-axis).




**Summary Of The Paper:**

This paper presents an approach to incorporate lattice priors into attention mask for deep learning. The theoretical results show this prior can be formulated as convolutions on an identity matrix. The proposed approach improves the sample efficiency of the model to solve the ARC tasks (few-shot geometric transformation learning) compared to vanilla transformers.


**Summary Of The Review:**

The idea of the paper is reasonable. Theoretical results and experiments on synthetic data show the proposed approach is superior to vanilla transformer. However, the experiments on real data seem relatively weak and it’d be valuable to see more evidences of LatFormer outperforming transformers and more existing approaches than SIFT/alpha-AMD on image registration benchmarks.

---

> ### Author Response · Authors · 2022-11-19
> **Reply to Reviewer ch2n**
>
> We thank the reviewer for the valuable feedback and for the time spent on our manuscript. We tried to improve the paper by incorporating the suggestions of all reviewers and we kindly ask to have a look at the general reply and the updated version of the paper. We answer your questions below.
>
> > It’d be valuable to show more experiments on real data. How about showing the results on standard image registration/correspondence learning benchmarks?
>
> As explained in our general reply, the experiments on the image registration task were meant to provide some insights on the applicability of our method to natural images, but the core results of our paper are the ones on geometric reasoning on ARC. Our work is meant to make a step towards enabling neural networks with the capability of performing fundamental geometric operations in a sample efficient way. We do not claim to reach state-of-the-art performance on image registration. We improved the presentation of our work to make this point clear and we believe that the experiments on synthetic tasks, ARC, and LARC are enough to support our claims. ARC is widely regarded as one of the most challenging benchmarks for AI as it requires complex reasoning, few-shot learning and broad generalization capabilities. We believe that this kind of benchmarks provide a controllable setup which helps diagnose how different algorithmic components affect performance on the reasoning aspect. An analysis on real data will become especially pertinent once the synthetic case can be convincingly tackled, but at present deep learning models are still far from this.
>
> > In Table 3, are there other baselines to compare with? I’d imagine a vanilla transformer would be a reasonable one.
>
> For the experiment on image registration, we used the same setup of Lu et al. (2021), with the same datasets and the same models. We performed an additional experiment with a vanilla transformer encoder as suggested, and then we recovered the transformation performed by the model from the attention matrix. The results are given in the table below and show that a vanilla transformer is not able to learn the task from just the few patches of an input and output image.
>
> |                           | **Aerial data** | **Cytological data** |
> |---------------------------|-----------------|----------------------|
> | **CycleGAN (A -> B)**     | 0.8             | 4.2                  |
> | **CycleGAN (B -> A)**     | 5.1             | 3.2                  |
> | **DRIT++ (A -> B)**       | 1.7             | 3.1                  |
> | **DRIT++ (B -> A)**       | 2.2             | 3.5                  |
> | **pixel2pixel (A -> B)**  | 2.1             | 2.2                  |
> | **$pixel2pixel (B -> A)** | 3.9             | 0.0                  |
> | **StarGAN (A -> B)**      | 4.3             | 1.7                  |
> | **StarGAN (B -> A)**      | 1.9             | 0.5                  |
> | **CoMIR**                 | 6.1             | 5.3                  |
>
>
> > In Figure 5, it seems that the transformer is barely learning the task. I’m curious how many samples would it take for a transformer to learn this task perfectly (how far should we extend the x-axis).
>
> We performed additional experiments with more examples and extended the x axis in Figure 5 from $2^{11}$ to $2^{14}$. However, Transformers and attention do not obtain good generalization performance, as these models still overfit the training data. As the number of examples increases, the whole set of experiments becomes computationally expensive and we are not able to extend the x axis further.

---

> > ### Author Response · Authors · 2022-11-28
> > **Reminder**
> >
> > Thanks again for the time spent on our manuscript and for your feedback. We clarified the scope of the paper and included additional experiments. Overall, we believe that the current version of the paper should address your initial concerns and we are looking forward to your feedback on our reply. Please let us know if you have any additional comments.

---

> > > ### Comment · Reviewer_ch2n · 2022-11-30
> > > **Thanks for the feedback.**
> > >
> > > Thank you for the feedback. After reading the author feedback and other reviews, I'm happy with the added experiments and clarification on the scope/contribution of this work. I have increased my score to 6.

---

### Official Review · Reviewer_vL2a · 2022-10-27

**Confidence:** 3
**Clarity, Quality, Novelty And Reproducibility:** The paper is clearly written. The nov…
**Correctness:** 2
**Technical Novelty And Significance:** 2
**Empirical Novelty And Significance:** 3
**Recommendation:** 5

**Strength And Weaknesses:**

Pros:
+ The idea of introducing lattice geometry and topology priors for attention design is interesting
+ The paper is clearly written.

Cons:
- The experiments on ARC and synthetic datasets and tasks are not sufficient to demonstrate the effectiveness and generalization of the design. It would be much more interesting if the method can be demonstrated on main-stream vision and NLP tasks to show the effectiveness of the designed attention mechanism.
- The ablation study should be extended to show the proposed attention mechanism with the lattice geometry and topology priors.


**Summary Of The Paper:**

In this paper, a LATFORMER model is introduced for learning effective attention. It incorporates both lattice geometry and topology priors for learning attention masks. The paper is demonstrated on ARC and on synthetic visual reasoning tasks, showing improved performances compared to standard attention and transformer on these tasks.

**Summary Of The Review:**

The experiments could be enhanced. More challenging tasks and datasets could be considered to better validate the proposed attention mechanism. The novelty is moderate.

---

> ### Author Response · Authors · 2022-11-19
> **Reply to Reviewer vL2a**
>
> We thank the reviewer for their time and their feedback. We believe that the updated version of the paper should address the main concerns of the reviewers and we kindly ask you to have a look at the paper and the general reply. We further answer your main comments below.
>
> **Experiments on synthetic datasets are not sufficient.**
> In short, we argue that experiments on synthetic tasks are by themselves meaningful and interesting for the progress of the field: ARC is widely regarded as one of the _most challenging benchmarks for AI today_. Further, currently, neural models achieve very low performance on ARC, and the state-of-the-art results are obtained by symbolic algorithms.
> Nowadays neural networks are expected to solve a wide variety of tasks, ranging from algorithmic reasoning, solving puzzles, playing games, combinatorial optimization, planning, and complex geometric reasoning. Over the past years there has been significant evidence that, though neural networks can easily tackle pattern-matching tasks with real-world data, they still struggle with many tasks requiring complex reasoning, few-shot learning and broad generalization (which ARC is designed to benchmark). ARC provides a controllable setup which helps diagnose how different algorithmic components affect performance on the reasoning aspect. An analysis on real data will become especially pertinent once the synthetic case can be convincingly tackled.
> We added more results on ARC and some experiments on LARC (a dataset that augments ARC with human-written natural-language descriptions of the tasks) in order to provide stronger baselines based on neural program synthesis and more evidence that the tasks are so challenging that even models with access to both the input-output grid and a natural language description of the task fail to perform well.
>
>
> **Novelty.** We believe soft masking of attention weights has not been considered in other works in the same way done in our paper. This has also been noticed by other reviewers: _“I find infusing these priors using soft attention masks novel and interesting”_ (**Reviewer TgJP**); _“The proposed method presents an interesting way to infuse inductive biases and knowledge priors in deep neural networks, which is an important open problem for next step's developments of AI and neural networks”_ (**Reviewer 9teR**); _“The paper is clear and the ideas appear to be novel”_ (**Reviewer ch2n**).

---

> > ### Author Response · Authors · 2022-11-28
> > **Reminder**
> >
> > Thanks again for the time spent on our manuscript and for your feedback. We clarified the scope of the paper and included additional experiments. Overall, we believe that the current version of the paper should address your initial concerns and we are looking forward to your feedback on our reply. Please let us know if you have any additional comments.

---

### Official Review · Reviewer_TgJP · 2022-10-29

**Confidence:** 2
**Correctness:** 3
**Technical Novelty And Significance:** 3
**Empirical Novelty And Significance:** 2
**Recommendation:** 6

**Clarity, Quality, Novelty And Reproducibility:**

Clarity:
- I can get the general idea of the paper but I was not able to follow the details without a relevant background in these problems.

Quality:
- I think the evaluation quality is somewhat weak since there are no real baseline competitors (e.g., neural-symbolic visual reasoning work). The paper only compares with end-to-end trainable models such as CNN and transformers.

Novelty:
- I find infusing these priors using soft attention masks novel and interesting.

Reproducibility:
- The supplementary material provides more implementation details of the architectures and theorem proof.

**Details Of Ethics Concerns:**

I don't find ethics concerns.

**Strength And Weaknesses:**

Strength:
+ Interesting idea of infusing lattice symmetry prior to a problem into an end-to-end trainable transformer model.
+ The progress on ARC datasets using deep learning models is promising.
+ The method is technically sound. The CNN-based mask prediction (Lattice Mask Expert) is interesting and supports compositions of multiple actions (e.g., reflection and rotation).

Weakness:
- From the examples in the paper and supplementary material, it seems that rotation is only 90 degrees. Not sure if this makes the problem trivial to solve (if we know the symmetric group of testing input data).
- No visual examples and results on the cross-modal image registration tasks. I am not familiar with the tasks so it's a bit hard for me to judge the significance of the results and how the results validate the proposed method.
- I wasn't able to follow some of the details in Section 3.3 and Section 4. But it may be because I don't have the background on related problems.

**Summary Of The Paper:**

The paper addresses problems that involve learning a geometric transformation (e.g., rotation) on the input data. Examples of the problem are Abstraction and Reasoning Corpus. The core idea of the proposed method is to inject lattice symmetry prior as soft masks to modulate the attention weights in a transformer. The paper shows that the various lattice symmetry actions can be realized with attention masks (such as translation, rotation, and reflection). The results on ARC tasks show that using the proposed method with infused lattice symmetry prior can perform well (Table 2 and Figure 5) and is sample-efficient.

**Summary Of The Review:**

I think this is an interesting paper that shows end-to-end trained models can perform well on abstract reasoning tasks. I don't know the practical implication of solving these lattice symmetry problems and how the ideas in the paper could be used to solve some other tasks. The paper did show an image registration task. But it does not provide sufficient details (e.g., are these only translations to rotation and translation? what's the task? what metrics were used to evaluate the success in Table 3?) for me to understand the significance. For registering images from different modalities, one should also try stronger descriptor-matching methods beyond SIFT, e.g., SuperGlue.
Given these concerns, I am thus on the fence about this paper but leaning slightly positive.

---

> ### Author Response · Authors · 2022-11-19
> **Reply to Reviewer TgJP**
>
> We thank the reviewer for the overall useful feedback and the time spent on our manuscript.  We improved the paper by incorporating the feedback of all reviewers and we believe we addressed initial concerns. Please see our general reply and the updated version of the paper. We further reply to your main comments below.
>
> **Quality of baseline competitors.** There has not been much work on applying end-to-end neural models on ARC because neural networks fail to achieve good performance on this kind of complex visual-reasoning problems. We believe that this is an additional strength of our method, which is the first approach showing that neural networks can tackle these complex tasks even with few examples if the proper inductive biases are taken into account. _We added more baselines to the paper, including attention with relative positional encodings, PixelCNN and augmenting the training grids with precomputed transformations._ In order to compare to even stronger baselines, we included _additional experiments on LARC_, a dataset that pairs ARC tasks with natural language descriptions.  _On this dataset, we compared to strong competitors based on neural program synthesis_ (the same baselines reported in the LARC paper), _showing that LatFormer reaches better results both with and without access to natural language_. This shows that LatFormer can learn geometric transformation better than neural methods equipped with a symbolic DSL.
>
> **Clarity.** We tried to make the main intuitions easy to grasp even without some relevant background by including examples in low dimensions where possible in Section 3. More examples are reported in Appendix D.
>
> **Rotation is only by multiples of 90 degrees.** We confirm that rotation is only by multiples of 90 degrees. This does not make the problem easy to solve for two reasons. First, we added an experiment that shows that a transformer with access to precomputed rotations is still less efficient than LatFormer. Second, rotations on ARC are often not around the center of the image (because of padding), and we address this problem by coupling a rotation expert with a translation expert so that the model can translate the image and rotate around the center. This means that the size of the symmetry group that our model operates on is much larger than just rotations by 90 degrees.
>
> **Details on the image-registration experiment.** We moved the experiment on image registration to the appendix and we included more details that should answer the questions of the reviewer. In short, we followed the experimental setup of Lu et al (2021) and compared the model to the same baselines in their work. A registration is considered successful if the relative registration error (i.e., the residual distance between the reference patch and the transformed patch after registration normalized by the height and width of the patch) is below 2%. This experiment is meant to show the applicability of our model beyond ARC. We did not compare to stronger descriptor-matching methods because this is out of the scope of our work. We explained this in Appendix B.3 and added a reference to SuperGlue. Our work is a step towards incorporating symmetry priors in neural networks in a way that can enable sample-efficient geometric reasoning and we believe that our experiments on ARC provide evidence that using soft attention masks is a promising technique to achieve this goal.

---

> > ### Author Response · Authors · 2022-11-28
> > **Reminder**
> >
> > Thanks again for the time spent on our manuscript and for your feedback. We clarified the scope of the paper and included additional experiments. Overall, we believe that the current version of the paper should address your initial concerns and we are looking forward to your feedback on our reply. Please let us know if you have any additional comments.

---

### Author Response · Authors · 2022-11-19
**General reply**

We thank all reviewers for their time and for their constructive feedback. We are glad that reviewers found our work  (and specifically the idea of modulating attention weights with soft attention masks) novel and well motivated.

Our main takeaway from the reviews pertains to how the initial version of the paper presented our results on image registration, with some reviewers criticizing the baselines of this experiment as not being SotA. We completely agree on this point and stress that **our paper does _not_ aim to build a SotA image registration method nor to claim that LatFormer is generally SotA in mainstream NLP and CV tasks**.

In contrast, **our work focuses on artificial reasoning (ARC and its variants)** and takes a step towards enabling neural networks with the capability of performing fundamental geometric operations in a sample efficient way---an important open problem. To emphasize this, we explicitly updated the introduction by stating: _"Our paper focuses on ARC and its variants. We see the extension of LatFormer to other tasks as a promising avenue for future research"_.

To better align our experimental section with our contributions, in the revised paper:
* **We moved the experiment on image registration to the appendix** and made it explicit that our experiment does not aim to establish SotA performance, adding references to more recent baselines (see Appendix B.3). These changes were taken to avoid future misunderstandings about the scope and contributions of this work.
* **We added three baselines to the ARC experiment**. Following and extending the reviewer recommendation, _we included further comparisons with i) attention modules with relative positional encodings, ii) PixelCNN, and iii) a data augmentation technique_ where a transformer is provided with precomputed transformations as part of its input, so that the model has to learn only what part of the input needs to be copied. The added baselines corroborate the usefulness of our method.
* **We included a new set of results on LARC (Language-complete ARC)** . LARC [1] is a recently introduced benchmark that pairs ARC tasks with natural-language descriptions generated by human participants. The authors of the original paper evaluated several neural program synthesis methods with access to both the natural language description and the input-output pairs of the tasks. _Encouragingly, we found that LatFormer outperforms program synthesis approaches that have access to additional modalities on tasks requiring geometric knowledge priors. We also built a version of our model that can take advantage of the natural language descriptions, obtaining some improvements with respect to a plain LatFormer model._

Overall, we genuinely believe that the paper has improved thanks to the feedback of reviewers and that *the current version addresses all initial concerns*. We would kindly like to encourage reviewers to consider the latest version and engage in discussion with us if they have additional questions.


[1] _Communicating Natural Programs to Humans and Machines_. Samuel Acquaviva, Yewen Pu, Marta Kryven, Theodoros Sechopoulos, Catherine Wong, Gabrielle E Ecanow, Maxwell Nye, Michael Henry Tessler, Joshua B. Tenenbaum. NeurIPS 2022 Track on Datasets and Benchmarks.

---

### Author Response · Authors · 2022-12-12
**Final comment**

We thank all reviewers for their time and the overall positive evaluation. We are pleased that **Reviewer ch2n** and **Reviewer 9teR** replied to our comments and were satisfied with our rebuttal. We are still waiting to hear from **Reviewer TgJP**, **Reviewer vL2a** and **Reviewer YP2T**. We spent a lot of effort on the rebuttal and we believe that the paper improved by incorporating feedback from the reviewers.

* **Reviewer TgJP**: the experiments on LARC and the comparison with more baselines (including approaches based on neural program synthesis) should address the comment on the quality of baseline competitors and the missing comparison with neuro-symbolic methods. We further included more details on the experiment on image registration, better explained the purpose of that experiment and moved it to the appendix to address the remaining concerns.
* **Reviewer vL2a**: we explained in the paper why we target synthetic tasks and why ARC is widely regarded as a very challenging benchmark. We added further experiments and baselines to support our claims.
* **Reviewer YPT2**: following your suggestion, we included in the paper a baseline that applies transformations to the training data as a form of augmentation, showing that LatFormer performs better. We included in our reply the results of using a vanilla Transformer for the experiment on image registration, showing that a vanilla Transformer struggles to achieve good performance.

Considering that **discussion between reviewers and authors is only enabled until today**, we would like to receive your comments on our work. Thanks again for your time.

---

### Decision · Program_Chairs · 2023-01-20

**Decision:**

Reject

**Justification For Why Not Higher Score:**

Given the borderline reviews for the paper I took the opportunity to read the paper in depth and see if I can find justification for passing the bar for acceptance at the conference. First of all, I agree with the reviewers comments that strong baselines are warranted and are of central importance to the experiments. I am somewhat concerned though that several experiments including baseline comparisons were presented during the rebuttal period and not provided at the time of submission. There are two reasons for my concerns: (1) not having a complete paper at submission time and doing necessary experiments after submission time presents an unfair advantage with respect to other papers and submissions. (2) new experiments presented during rebuttals can not be as thoroughly vetted as other experiments.

With regard to the issues surfaced in (1) and (2), I have several concerns mirroring several of the reviewer comments:
* Given that the ARC results are central to the paper, it is critical to see more discussion about previous work on the ARC dataset, and how these results compare against previously presented work. I see minimal discussion about previous approaches, and the successes and limitations of the previous methods. This seems critical for placing the quality and strength of the results in context. For instance, there was a [Kaggle competition](https://www.kaggle.com/competitions/abstraction-and-reasoning-challenge/) 3 years ago on the ARC dataset which included almost 1000 submissions. How do the numbers presented compare with those reported on their leaderboard? What methods did these submissions use? Did the authors benchmark against these methods? Why or why not?
* I recognize that the image registration experiments are not considered central to the paper by the authors, however showing some potential applicability of the proposed method to real world problems is an important criteria for judging the quality of the proposed methods. That is, a synthetic task serves us a useful purpose in our field to provide a probe or proxy for a more difficult problem that is hard to measure or assess. If the authors are not able to demonstrate the application of their method to a real world problem, then it is incumbent on the authors to provide ready discussion and citations arguing why performing better on the synthetic task (e.g. ARC) would lead to better performance on real world tasks.
* A discussion about how a synthetic task provides a valuable proxy for real world tasks does not remove the responsibility of authors to attempt to demonstrate the applicability of their method on a real world task. So long as the method may be readily applied to a real world problem, it is incumbent upon the authors to make a good faith effort to test out their method on this task. I believe that several of the reviewers implicitly reflected this sentiment, and would rather see the results on image registration not be demoted in the manuscript. Accordingly, it is important to show solid results on this real world task that this method is amenable to, and if not, be able to provide some explanation for the limitations of the method. Delving into these experiments, I see several items of concern remaining: Namely, the selection of using SIFT as a baseline appears quite dated. SIFT was a mainstay of computer vision before the advent of deep learning and has been superseded by many methods. For instance, in a simple search I just performed I see many methods employing deep learning that were not compared against ([Deep Image Homography Estimation](https://arxiv.org/abs/1606.03798) (2014); [Multi-Temporal Remote Sensing Image Registration Using Deep Convolutional Features](https://ieeexplore.ieee.org/document/8404075) (2018); [COTR: Correspondence transformer for matching across images](https://arxiv.org/abs/2103.14167) (2021)). The selection of the selected baselines need to be better justified and I would like to see the authors compare against more modern methods.

Given these issues, I think more work is needed to properly describe the baselines and experiments the authors investigated and place these in the context of previous work. Additionally, I would suggest that all of these results be incorporated into the main body of the paper. For these reasons, this paper will not be accepted into this conference at this time. I suspect that given all of these recent experiments, this is quite achievable and a reformulated version of this paper would make a good submission to a future machine learning venue.


**Justification For Why Not Lower Score:**

n/a

**Metareview: Summary, Strengths And Weaknesses:**

In this work, the authors propose a new method for learning a geometric transformation to aid problems of abstraction and reasoning. In particular, the authors propose the addition of a lattice symmetry prior to adjust the attention weights of a Transformer. The authors first provide some theoretical results showing proof of existence and use these theoretical results to demonstrate that modifications to a standard attention layer through a mask generated by a convolutional network are sufficient to achieve a geometric prior. The authors demonstrate the efficacy of their results on the ARC and LARC datasets (i.e. synthetic visual benchmarks) providing good performance in a sample efficient manner. In particular, the authors show that their method may achieve comparable results to standard self-attention and Transformers but with two orders of magnitude fewer data.

The reviewers commented positively on the novelty of the idea, the empirical success on the ARC dataset, and the technical soundness of the approach. The reviewers did express some concerns about the generality of the approach, the clarity of some of the sections and the lack of qualitative examples. The two largest concerns though were (1) the lack of strong baselines and (2) the connection to real world problems.

Addressing (1), the authors responded with additional baselines showcasing the performance of models with attention as well as auto-regressive models (e.g. PixelCNN). In addition, the authors performed notable ablations with data augmentations to see if such a simple suggestion might suffice to match their proposed method. Their proposed method outperformed all of these benchmark comparisons indicating that their method provided non-trivial gains over standard methods. Addressing (2), the authors took the reviewer’s suggestions and benchmarked their method on an image registration/correspondence task. The authors performed additional experiments on image registration which they include in the Appendix. The authors compared their method against SIFT and alpha-AMD. Their method again showcased superior performance.

**Summary Of Ac-Reviewer Meeting:**

I attempted to organize a meeting but I was not able to find a timely meeting given the challenges of time zone and everyone's timing.